# Integrated cooling (i-Cool) textile of heat conduction and sweat transportation for personal perspiration management

Yucan Peng [1,9], Wei Li [2,3,9], Bofei Liu[1], Weiliang Jin [2], Joseph Schaadt[4,5], Jing Tang[1], Guangmin Zhou[1], Guanyang Wang[6], Jiawei Zhou [1], Chi Zhang [7], Yangying Zhu[1], Wenxiao Huang[1], Tong Wu[1], Kenneth E. Goodson[7], Chris Dames[4,5], Ravi Prasher [4,5], Shanhui Fan [2] & Yi Cui [1,8 ✉]

Perspiration evaporation plays an indispensable role in human body heat dissipation. However, conventional textiles tend to focus on sweat removal and pay little attention to the basic thermoregulation function of sweat, showing limited evaporation ability and cooling efficiency in moderate/profuse perspiration scenarios. Here, we propose an integrated cooling (i-Cool) textile with unique functional structure design for personal perspiration management. By integrating heat conductive pathways and water transport channels decently, i-Cool exhibits enhanced evaporation ability and high sweat evaporative cooling efficiency, not merely liquid sweat wicking function. In the steady-state evaporation test, compared to cotton, up to over 100% reduction in water mass gain ratio, and 3 times higher skin power density increment for every unit of sweat evaporation are demonstrated. Besides, i-Cool shows about 3 °C cooling effect with greatly reduced sweat consumption than cotton in the artificial sweating skin test. The practical application feasibility of i-Cool design principles is well validated based on commercial fabrics. Owing to its exceptional personal perspiration management performance, we expect the i-Cool concept can provide promising design guidelines for next-generation perspiration management textiles.

[1] Department of Materials Science and Engineering, Stanford University, Stanford, CA, USA. [2] E. L. Ginzton Laboratory, Department of Electrical Engineering, Stanford University, Stanford, CA, USA. [3] GPL Photonics Lab, State Key Laboratory of Applied Optics, Changchun Institute of Optics, Fine Mechanics and Physics, Chinese Academy of Sciences, Changchun, China. [4] Department of Mechanical Engineering, University of California, Berkeley, CA, USA. [5] Energy Technologies Area, Lawrence Berkeley National Laboratory, Berkeley, CA, USA. [6] Department of Mathematics, Stanford University, Stanford, CA, USA. [7] Department of Mechanical Engineering, Stanford University, Stanford, CA, USA. [8] Stanford Institute for Materials and Energy Sciences, SLAC National Accelerator Laboratory, Menlo Park, CA, USA. [9] These authors contributed equally: Yucan Peng, Wei Li. ✉email: yicui@stanford.edu

Satisfaction with the thermal environment for human body is significant, not merely due to the demand for comfort, but more importantly because thermal conditions are crucial for human body health[1]. Heat-resulted physiological and psychological problems not only can be threatening for human health[2], but also negatively influence labor productivity and society economy[3]. Personal thermal management focusing on thermal conditions of human body and its local environment is emerging as an energy-efficient and cost-effective solution[4,5]. Without consuming excess energy on managing the temperature of the entire environment[6,7], innovative textiles have been designed for controlling human body heat dissipation routes[8,9]. In general, human body dissipates heat via four different pathways: radiation, convection, conduction and evaporation[10]. Recently, textiles with engineered radiative properties[11–16], convective and conductive properties[17–19] have been demonstrated as promising approaches for personal thermal management especially for mild scenarios. However, for intense scenarios, textiles for ideal personal perspiration or evaporation management are still lacking.

For the delicate human body system with a narrow temperature range (36–38 °C core temperature at rest and up to 41 °C for heavy exercise)[20], evaporation plays an indispensable role in human body thermoregulation. Even at a mild state, about 20% of heat dissipation of the dry human body relies on the water vapor loss via insensible perspiration[10,21]. With further increase of heat load, liquid sweat evaporation contributes to more and more heat loss and becomes the major route for human body heat dissipation in intense scenarios such as heavy exercise and hot/humid environments, where excess heat cannot be dissipated efficiently by other pathways[22,23]. State-of-the-art textiles for daily use are usually sufficiently good at water vapor transmission to ensure comfort at the mild state (See Supplementary Note 1 and Supplementary Figs. 1–2 for more discussion)[24]. Nevertheless, the cooling performance of conventional textiles is to be improved when human body is in more intense scenarios, such as moderate/profuse perspiration situations in which liquid sweat is inevitably present.

In order to avoid increased wettness on the skin which causes less comfort in such cases[25,26], state-of-the-art textiles, including moisture management fabrics, tend to focus on sweat removal. Textiles made of natural fibers, such as cotton, show strong water absorption capacity, which can help alleviate sense of wettness quickly[27]. In spite of diminished absorbing ability, synthetic fibers (with profiled cross-section), such as polyester, are developed to possess enhanced moisture transportation than natural fibers to deliver water to the textile surface for faster evaporation[28,29]. Microfibres are also explored for improved wicking[30]. Besides, strategies including surface hydrophilicity/hydrophobicity modification[31–33], multiple-layer design with differential wettability[34,35] and hierarchical design of multiscale interconnected pores with capillarity gradient[36,37] are reported to realize better controlled directional water transportation. These textiles serve as a buffer absorbing water to provide dry sense for people and can potentially offer a comparatively larger surface area for evaporation.

However, how to efficiently unlock the cooling power of sweat evaporation for human body thermoregulation and design textiles based on laws of human body perspiration process have not been taken into account. In the aspect of thermoregulation, sweat is secreted to be evaporated and take away the excess heat. Nevertheless, although sweat evaporation does happen on the conventional textiles, human skin underneath is not effectively cooled since heat for vaporization is not efficiently drawn from the skin because of the limited heat transfer[38–40]. One extreme case is that only the textile surface rather than human skin can be cooled. In other words, the sweat absorbed in the conventional textiles

shows decreased evaporative cooling efficiency in cooling the human body, which means sweat is less efficiently utilized. Also, even regarding evaporation rate of conventional textiles, it is relatively restrained because skin heat cannot be efficiently delivered to the evaporation interface to accelerate evaporation. The inefficient cooling effect will lead to further perspiration, and meanwhile the slow sweat evaporation, will result in the accumulation of sweat in the textile. This process may undermine the buffer effect of the textiles once the absorption limit of the fabric is reached, at which point the human body will get wet and sticky again. The excessive perspiration can also cause potential risk of dehydration, electrolyte disorder, physical and mental deterioration or even death[41]. Moreover, when people are in highly active scenarios, the maximum cooling power of sweat evaporation that can be achieved actually limits the maximum activity level of human body[42]. Accordingly, in addition to decent wicking property, an optimal textile for perspiration scenario should show high evaporation ability and more importantly high sweat evaporative cooling efficiency to utilize sweat in a highly efficient manner, to provide adequate cooling effect using minimized amount of sweat.

In this work, we propose a novel concept of integrated cooling (i-Cool) textile of heat conduction and sweat transportation to achieve the as-mentioned goals based on human body perspiration process, as illustrated in Fig. 1a. We introduce heat conductive components into the textile and divide the functionalities of heat conduction and sweat transport into two operational components. The heat conductive matrix and sweat transportation channels are integrated together in the i-Cool textile. The synergistic effect of the two components results in excellent performance at sweat wicking, fast evaporation, efficient evaporative cooling for human body and reducing human body dehydration. As shown in Fig. 1b, the sweat transport channels can pull liquid water up from skin and spread it out in the sweat transport channels for evaporation. On the other hand, the heat conductive matrix can efficiently transfer skin heat to the evaporation spots that are integrated on the heat conductive matrix[43,44]. Therefore, combined with large evaporation area and efficient heat conduction from skin, sweat absorbed in the water transportation channels can be evaporated quickly into air, taking away a huge amount of heat from the skin. The efficient heat removal from the skin provides improved evaporative cooling effect and decrease skin temperature effectively, which will consequently reduce human body dehydration. As illustrated in Fig. 1c, compared to the conventional textiles, the i-Cool textile functions not only to wick sweat but also provide heat conduction paths for the accelerated evaporation and efficiently take away a great amount of heat from the skin. Furthermore, the enhanced evaporation ability and high sweat evaporative cooling efficiency can prevent the i-Cool textile from flooding to a much greater extent and avoid excessive perspiration. The improved evaporative cooling effect does not mean more sweat needs to be generated or even evaporated. Therefore, the i-Cool textile can help human body achieve enhanced cooling effect with greatly reduced sweat secretion by using the sweat in a highly efficient manner.

## Results and discussion
On the basis of the i-Cool functional structure design principles as outlined above, we selected copper (Cu) and nylon 6 nanofibres for proof of concept. It is worthwhile to mention that Cu and nylon 6 nanofibres are not the only choices. Other materials satisfying the design principles can be applied as well. Here, Cu is well-known for its extraordinary thermal conductivity (~400 $W \cdot m^{-1} \cdot K^{-1}$), and nylon 6 nanofibres are capable of water wicking. As illustrated in Supplementary Fig. 3, electrospinning

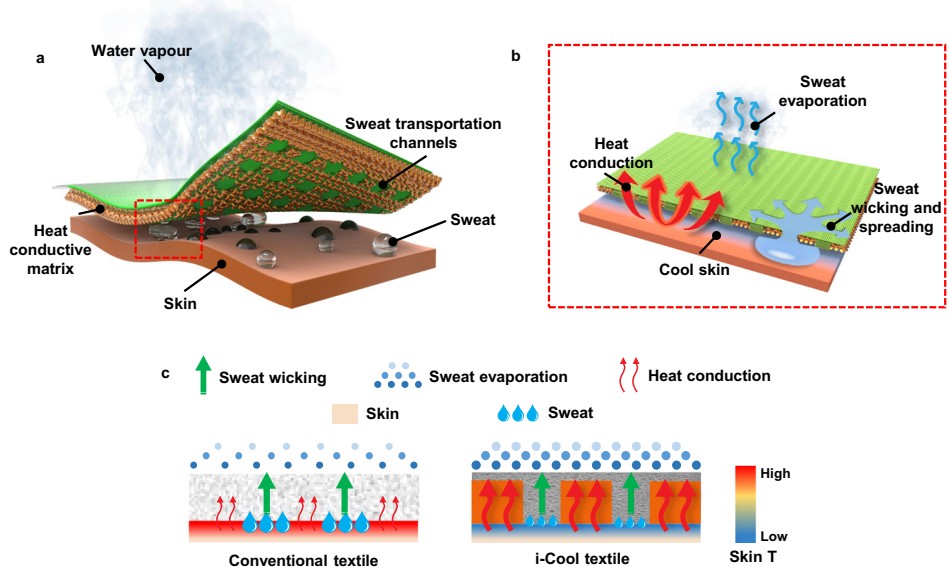

**Fig. 1 Schematic of the functional structure design of integrated cooling (i-Cool) textile of heat conduction and sweat transportation for personal perspiration management and its working mechanism. a**, Schematic of the i-Cool textile. The synergistic effect of the heat conductive matrix and sweat transport channels provides a solution to textile in personal perspiration management. **b**, Schematic of the working mechanism of the i-Cool textile. When human body perspires, the water transport channels can wick sweat from the skin surface and spread sweat onto the large-area top surface made of fibers quickly. The heat conductive matrix transfers human body heat efficiently to where the evaporation happens, to assist fast evaporation. Meanwhile, it can deliver the evaporative cooling effect to human body skin efficiently. **c**, Comparison between conventional textiles and the i-Cool textile. Conventional textiles usually offer comfort via buffer effect of absorbing sweat, which is helpful to relieve discomfort of wet and sticky sense. However, its limited evaporation rate and evaporative cooling efficiency cannot provide effective cooling effect for skin and may undermine the buffer effect soon. Different from normal textiles, the i-Cool textile functions not only to transport sweat but also provide an excellent heat conduction path for the accelerated evaporation and taking away a great amount of heat from the skin, which can prevent the i-Cool textile from flooding to a much greater extent and avoid excessive perspiration. Therefore, the i-Cool textile can help human body achieve enhanced cooling effect with greatly reduced sweat, by using the sweat in a highly efficient manner. The weight contrast in red arrows drawing illustrates the heat transport ability difference. The dot size and density contrast in the sweat evaporation drawing shows the different evaporation ability. The drop size contrast in the sweat drawing illustrates that i-Cool textile can help reduce sweat consumption.

was utilized to generate nylon 6 nanofibres, which were transferred to the heat conductive Cu matrix prepared by laser cutting. With press lamination, the i-Cool (Cu) textile with desired functional structure design was fabricated. The photograph of as-fabricated i-Cool (Cu) textile is displayed in Fig. 2a. Nylon 6 nanofibres not only cover the Cu top surface, but also fill inside the pores, as shown in the magnified photograph of the bottom side of the i-Cool (Cu) textile in the inset of Fig. 2b. Nanofibres on the skeleton of Cu matrix are denser with smaller void space among the nanofibres than the ones in the pores of Cu matrix, which can be clearly observed in the scanning electron microscope (SEM) images in Fig. 2b and Supplementary Fig. 4. The capillary difference resulted from the morphology difference benefits one-way directional water transportation from inner surface to outer surface. To evaluate the performance of the i-Cool (Cu) textile, we selected cotton textile as the main control textile since it is arguably the most widely used and accepted textile in human history. We have also chosen other well-known activewear fabrics for comparison purposes.

**Liquid water transport characterization**. Textiles designed for perspiration scenarios must be able to wick sweat from the skin (in contact with textile bottom) and spread it out. Correspondingly, we tested in parallel the i-Cool (Cu) textile and commercial textiles including cotton, Dri-FIT, CoolMax and Coolswitch via mimicking the sweat transport process from the human body skin to the outer surface of the textile. Textile samples covered a certain amount of liquid water on the platform respectively, and

the wicking rate was calculated via dividing wicking area by wicking time for every sample (Supplementary Fig. 5). It turned out that the interconnected nylon 6 nanofibres in the i-Cool (Cu) textile was able to quickly transport liquid water from bottom to top and spread it out, which exhibited comparable or higher wicking rate in comparison with conventional textiles (Fig. 2c). Besides, due to the unique structure design and the nanofibre morphology variation from i-Cool (Cu) bottom to the outer surface, i-Cool (Cu) exhibits good one-way water transport property. As displayed in Supplementary Fig. 6a, the water droplet added onto the inner side of i-Cool (Cu) can be transported to the outer surface and spread out very quickly while little water remained on the inner side. In reverse, water transportation was limited when the water droplet added to the outer side. As a comparison, for cotton, the testing time on the outer side and inner side was almost the same no matter which side the water droplet was added onto (Supplementary Fig. 6b), which means the conventional cotton fabric shows no one-way transport capability. Also, in the scenario of adding water onto inner side, the water spreading rate on the inner surface and outer surface ($S_{inner}$ and $S_{outer}$) and one-way transport index ($\mu$) were defined (See Methods for more details) and plotted in Supplementary Fig. 7[45]. The i-Cool (Cu) shows obviously different $S_{inner}$ and $S_{outer}$, and very large $\mu$, while $S_{inner}$ and $S_{outer}$ are very similar for cotton and its $\mu$ is very close to 1, which demonstrates the apparent one-way sweat transport advantage of i-Cool (Cu) again. This property can also help faster evaporation, because sweat can spread on the outer surface quickly and liquid water transport to the nanofibres right on the heat conductive Cu matrix is preferential[37].

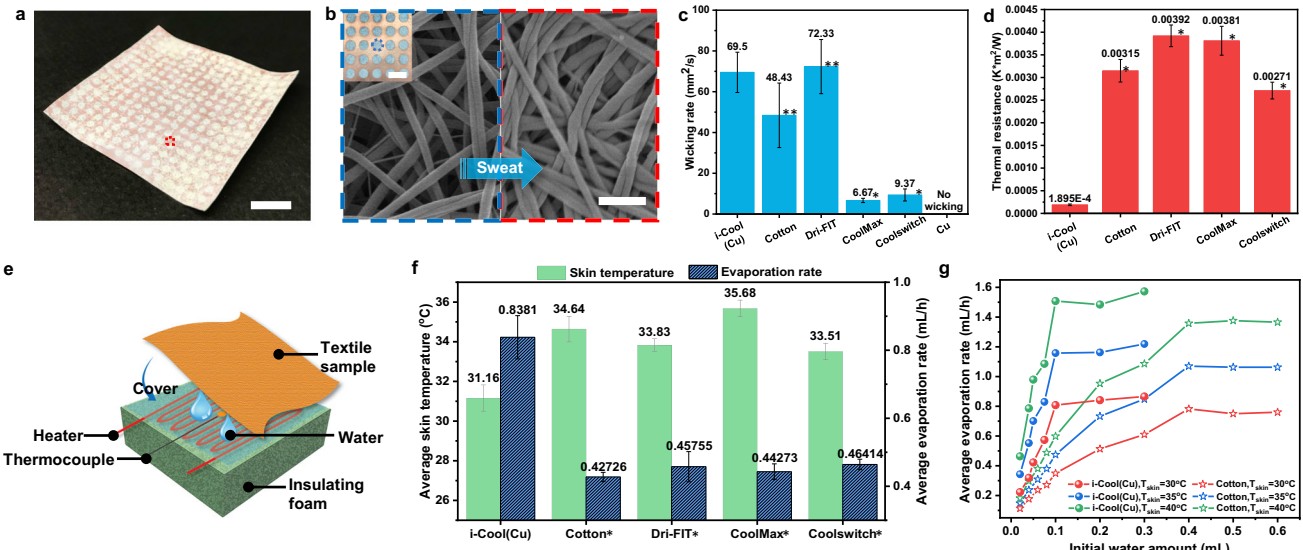

**Fig. 2 Wicking performance, thermal resistance and transient droplet evaporation test of the i-Cool (Cu) textile. a**, Photograph of as-prepared i-Cool (Cu) textile. Scale bar, 1 cm. **b**, SEM image of nylon 6 nanofibres in the pores of heat conductive matrix (blue dash box) and on the top of heat conductive matrix skeleton (red dash box). Sweat tends to be transported to the nanofibres on the heat conductive matrix skeleton due to the morphology difference. Scale bar, 1 μm. Inset is the magnified photograph of the bottom side of i-Cool (Cu) textile showing its integrated heat conduction channels and water transport channels. The holes are 2 mm in diameter and 3 mm pitch. Scale bar, 4 mm. **c**, Wicking rate of the i-Cool (Cu), cotton and other commercial textiles. It shows how fast water underneath the textile can be pulled up and spread on the top surface. Double asterisks, Statistical significance between the i-Cool (Cu) and labeled sample, Welch's $t$-test $p < 0.1$; Asterisk, Statistical significance between the i-Cool (Cu) and labeled sample, Welch's $t$-test $p <$ 0.001. **d**, Thermal resistance of the i-Cool (Cu), cotton and other commercial textiles measured by cut-bar method (See more discussion in Supplementary Note 2). Asterisk, Statistical significance between the i-Cool (Cu) and labeled sample, Welch's $t$-test $p < 0.001$. **e**, Schematic illustration of the transient droplet evaporation test. **f**, Average skin temperature and average evaporation rate of the i-Cool (Cu) textile and the conventional textiles (initial water amount: 0.1 mL, skin heater power density: 422.5 W/m$^2$). Asterisk, Statistical significance of average skin temperature between the i-Cool (Cu) textile and other textile samples, Welch's $t$-test $p < 0.001$. Statistical significance of average evaporation rate between the i- Cool (Cu) textile and other textile samples, Welch's test $p < 0.001$. **g**, Fitted average evaporation rate of i-Cool (Cu) and cotton versus initial water amount at different skin temperature. All the error bars represent standard deviation of measured data.

**Thermal resistance measurement**. To quantify the enhancement of heat transport capability of the i-Cool (Cu) textile, we performed the measurement of thermal resistance using cut bar method, as illustrated in Supplementary Fig. 8. Using this method, we measured the dry thermal resistance of the i-Cool (Cu) textile and other commercial textile samples all under an additional contact pressure of ~15 psi (103 kPa). As exhibited in Fig. 2d, the i-Cool (Cu) textile shows about 14–20 times lower thermal resistance compared to the conventional textiles (See Supplementary Note 2 and Supplementary Fig. 8 for more details and discussion). A thermal resistor model was built up to interpret the measured thermal resistance. It was found out the nylon 6 nanofibre layer contributes to the major thermal resistance, and increasing the thickness of heat conductive matrix (Cu) will only cause minor increase of thermal resistance (Supplementary Fig. 9). It provides support for the possibility of extending the i-Cool concept into fabrics of various thickness.

**Transient droplet evaporation test**. We further used a transient droplet evaporation test to compare the evaporation performance of the i-Cool (Cu) textile and the conventional textiles. Figure 2e illustrates the experimental setup: A heater placed on an insulating foam was used to simulate human skin with a thermocouple attached to the heater surface; We added liquid water at 37 °C to mimic sweat onto the artificial skin, then textile samples covered on the wet artificial skin immediately; The power density of the artificial skin was maintained constant during the measurement. During the whole evaporation process, skin temperature was always monitored and recorded. For example, a group of typical curves of skin temperature versus time are shown in

Supplementary Fig. 10. Generally, the curves can be divided into three stages for every tested textile sample. Initially, when water was just added onto the artificial skin, skin temperature dropped sharply. Then, skin temperature was relatively stable only fluctuating in a small range in the evaporation stage. Eventually, skin temperature rose again quickly once water was completely evaporated.

Two pieces of important information can be obtained through comparing the curves of i-Cool (Cu) and the conventional textiles. Firstly, the evaporation time with i-Cool (Cu) was much shorter, which indicates that i-Cool (Cu) exhibits higher evaporation rate. This conclusion can also be verified by measuring the mass loss of liquid water over time during the evaporation test (Supplementary Fig. 11). Secondly, skin temperature with i-Cool (Cu) textile was lower than the conventional textiles during evaporation, demonstrating human body can evaporate sweat faster with even lower skin temperature when a person wears i-Cool textile. The summarized comparison of average skin temperature and average evaporation rate between the i-Cool (Cu) textile and the conventional textiles is displayed in Fig. 2f (0.1 mL initial water, 422.5 W/m$^2$ power density, ambient temperature: ~22 °C). The i-Cool (Cu) shows 2.3–4.5 °C lower average skin temperature and about twice faster average evaporation rate compared to the conventional textiles.

Furthermore, measurements under assorted skin power density and initial liquid water amount for i-Cool (Cu) and cotton were performed. With different experimental parameters, the average evaporation rate was calculated and plotted versus the average skin temperature during evaporation in Supplementary Fig. 12a and Supplementary Fig. 12b. In our measurement range, a linear

relationship between the average evaporation rate and the average skin temperature was observed with a certain amount of initial water. Employed the linear fitting relationship and replotted from Supplementary Fig. 12, Fig. 2g shows the fitted relationship between the average evaporation rate and the initial water amount at different skin temperatures for the i-Cool (Cu) and cotton. Generally, the average evaporation rate increases as the initial water amount increases and it shows an approaching saturation trend as the initial water amount reaches a certain level. This is perhaps consistent with the change trend of average evaporation area during the drying process when the initial water amount is changed. It is obvious that the i-Cool (Cu) exhibits overall higher evaporation rate than cotton. Besides, i-Cool (Cu) can achieve this with lower initial water amount and lower skin temperature, indicating the superiority in sweat evaporation of the i-Cool functional structure design.

**Steady-state evaporation test**. In order to further characterize the evaporation features of i-Cool (Cu) and analyze its advantages over conventional textiles, we performed a steady-state evaporation test. Compared to the transient droplet evaporation test above, the steady-state evaporation test can help derive additional useful indexes to differentiate the evaporation property of textiles during human body perspiration. The measurement apparatus is illustrated in Fig. 3a. Similarly, a heater placed on an insulating foam was used to simulate human skin. Thermocouples and a water inlet which were sealed in a thin acrylic board were attached to the artificial skin surface. Not adding a certain initial amount of water, water heated to 37 °C was pumped onto the skin surface at a specific rate continuously, and textiles on it wicked the intake water. Power density of the skin was adjusted to maintain skin temperature stable at 35 °C. The system with textile samples finally reached a steady-state. By changing steady-state evaporation rate (i. e. water pumping rate), the corresponding stable water mass gain and power density can be measured for different textiles.

Figure 3b exhibits the measured water mass gain ratio (i. e. water mass gain/textile sample dry mass*100%, denoted as $W$) of i-Cool (Cu), cotton and Dri-FIT versus increasing evaporation rate (denoted as $v$). Firstly, it was observed that the water mass gain ratio of i-Cool (Cu) was always lower than cotton and Dri-FIT at the same evaporation rate, indicating that less sweat is required to "activate" i-Cool (Cu) to reach the same evaporation rate compared to the conventional ones. For example, when the steady-state evaporation rate was 1.1 mL/h, i-Cool (Cu) only showed about 20 percent of water mass gain ratio, while $W$ of cotton was approximately 130 percent. This phenomenon was also in accordance with the transient droplet evaporation test results. Furthermore, we fitted the curves in Fig. 3b and calculated water mass gain ratio gradient ($dW/dv$), as shown in Fig. 3c. $dW/dv$ of i-Cool (Cu) is apparently smaller than the conventional textiles, even if all of them displayed water mass gain increase as the growth of evaporation rate. Besides, $dW/dv$ of cotton and Dri-FIT rises rapidly with the increase of evaporation rate, especially cotton. It means that it becomes even more and more difficult to achieve higher evaporation rate. Nevertheless, this index for i-Cool (Cu) stays almost unchanged in the measurement range. During real human body perspiration, these features of i-Cool (Cu) enables it to fast evaporate sweat before sweat accumulates a lot and to retain a relatively dry state even during very profuse perspiration that requires high evaporation rate.

The measured power density (denoted as $q$) of artificial skin in this test is shown in Fig. 3d. Overall, the skin power density with i-Cool (Cu) was higher than the conventional textiles when they were at the same evaporation rate, demonstrating the cooling ability of i-Cool (Cu) during perspiration is stronger. It is worthwhile to mention that i-Cool (Cu) is easier to reach higher

evaporation rate, thus the cooling power difference between i-Cool (Cu) and conventional textiles in practical use can be further enlarged. Besides, the curves in Fig. 3d were fitted and power density gradient ($dq/dv$) could be derived, as displayed in Fig. 3e. This index ($dq/dv$) exhibits the cooling power increment rate when evaporation rate increases. Obviously, $dq/dv$ of i-Cool (Cu) is much higher than cotton and Dri-FIT, which means i-Cool (Cu) can provide much higher cooling power when every unit of sweat evaporates. To be specific, $dq/dv$ of i-Cool (Cu) is about 3 times higher than that of cotton and Dri-FIT. Furthermore, to some extent, $dq/dv$ can be converted into sweat evaporative cooling efficiency (denoted as $\eta$) (See Supplementary Note 3 for more discussion). Based on our estimation, the evaporative cooling efficiency of i-Cool (Cu) is 0.8~1, while $\eta$ of cotton and Dri-FIT is only 0.2~0.4 (Supplementary Fig. 13). Therefore, we demonstrated i-Cool (Cu) shows evident advantages in both evaporation ability and sweat evaporative cooling efficiency, which makes it to be promising in next-generation textiles for personal perspiration management.

**Artificial sweating skin platform with feedback control loop**. Human body is capable of adjusting itself to maintain home-ostasis in the means of feedback control loops[46]. Taking per-spiration as an example, when the human body temperature exceeds a threshold, the sympathetic nervous system stimulates the eccrine sweat glands to secrete water to the skin surface. In reverse, water evaporation on the skin surface accelerates heat loss and thus body temperature decreases, which will reduce or suspend the perspiration of human body (Fig. 4a)[47,48].

To mimic human body perspiration situation and show the performance difference between the i-Cool (Cu) textile and the conventional textiles, we designed an artificial sweating skin platform with feedback control loop, as illustrated in Fig. 4b. In this system, an artificial sweating skin that can generate sweat uniformly from every fabricated perspiration spot was built up and served as the test platform. Power was supplied to the artificial sweating skin platform to generate heat flux simulating human body metabolic heat. A syringe pump and a temperature controller were utilized to provide continuous liquid water supply at a constant temperature (37 °C) for the artificial sweating skin. A thermocouple was attached to the artificial sweating skin platform surface, monitoring skin temperature with a thermo-couple meter that transmitted skin temperature data to the computer in real time. Subsequently, the internal set program could instantly alternate the pumping rate of the syringe pump that corresponds to the sweating rate of artificial sweating skin, which realized the feedback control loop imitating human body's feedback control mechanism.

To achieve uniform water outflow through each artificial sweat pore mimicking human body skin sweating, we designed the artificial sweating skin platform as illustrated in Fig. 4c. In the bottom, an enclosed small cuboid cavity connecting to water inlet acted as a water reservoir. When water was pumped in, water in the reservoir was forced out upwards through the channels on the reservoir cap. On the top of it, a perforated hydrophilic heater was attached to generate heat, in the meantime through which water can flow out. The uniform "sweating" from each artificial sweat pore was realized by the fabricated Janus-type wicking layer with limited water outlets that was placed above the perforated heater (See Supplementary Note 4–5 and Supplementary Figs. 14–16 for more details and discussion).

We believe that the measurement results obtained with the as-built artificial sweating skin platform can provide reasonable parallel thermal comparison among the textile samples, even though this set-up cannot fully represent the human body due to the lack of some other feedback control mechanisms such as

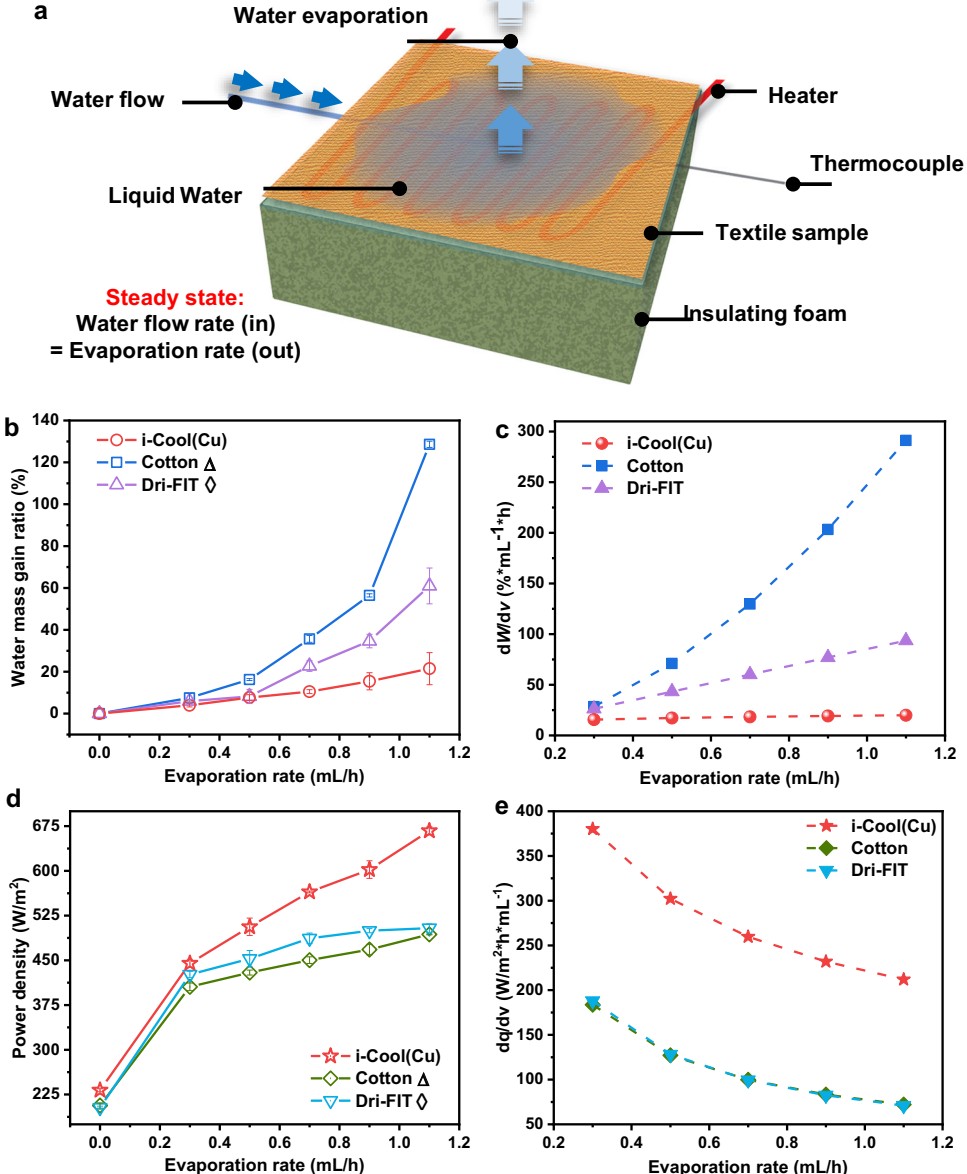

**Fig. 3 Steady-state evaporation test of the i-Cool (Cu) textile, cotton and Dri-FIT. a,** Schematic illustration of the measurement apparatus and method. **b,** Measured water mass gain ratio ($W$) at different evaporation rate ($v$). Triangle, Statistical significance between the i-Cool (Cu) and cotton, Welch's $t$-test $p < 0.1$ at 0.3 mL/h, $p < 0.001$ at 0.7 mL/h, $p < 0.01$ for others. Diamond, Statistical significance between the i-Cool (Cu) and Dri-FIT, Welch's test $p < 0.05$ at 0.3 mL/h, no statistical significance at 0.5 mL/h, $p < 0.01$ for others. **c,** $dW/dv$ obtained by fitting data in (b). i-Cool (Cu) can achieve a certain evaporation rate with much lower water gain. The required water gain increase for larger evaporation rate is also reduced. **d,** Measured power density ($q$) at different evaporation rate ($v$). Triangle, Statistical significance between the i-Cool (Cu) and cotton, Welch's $t$-test $p < 0.05$ at 0.3 mL/h, $p < 0.001$ at 0.7 mL/h, 0.9 mL/h, $p < 0.01$ for others. Diamond, Statistical significance between the i-Cool (Cu) and Dri-FIT, Welch's test shows no statistical significance at 0.3 mL/h, $p < 0.05$ at 0.5 mL/h, $p < 0.01$ at 0.7 mL/h, 0.9 mL/h, $p < 0.001$ for others. **e,** $dq/dv$ obtained by fitting data in (d). The i-Cool (Cu) can show enhanced cooling effect with higher sweat evaporative cooling efficiency. All the error bars represent standard deviation of measured data.

blood flow feedback control and the differences in size, shape, thermal capacity, etc. With the realization of scale-up, we expect to conduct the human physiological wear experiment[42] in the near future.

**Artificial sweating skin test.** On the artificial sweating skin platform, we first performed a demonstrative experiment to intuitively show the sweat evaporative cooling efficiency difference. In this experiment, the same power density was used for the i-Cool (Cu) textile and cotton textile while the sweating rate was varied for different ones to realize the same skin temperature

(34.5 °C), then we observed the condition of the artificial skin device and the textile samples after stabilization of 30 minutes. As shown in Supplementary Fig. 17, bare skin remained almost dry. The skin with the i-Cool (Cu) textile also remained dry while there was a little water absorbed in the sample. Nevertheless, there was a much larger amount of water remaining on both the skin platform and the cotton textile. These results intuitively demonstrated the i-Cool (Cu) can cool down the skin more efficiently consuming much less sweat.

Then, we performed measurements with constant skin power density for i-Cool (Cu) and other commercial textile samples, to mimic an exercise scenario of human body (See Supplementary

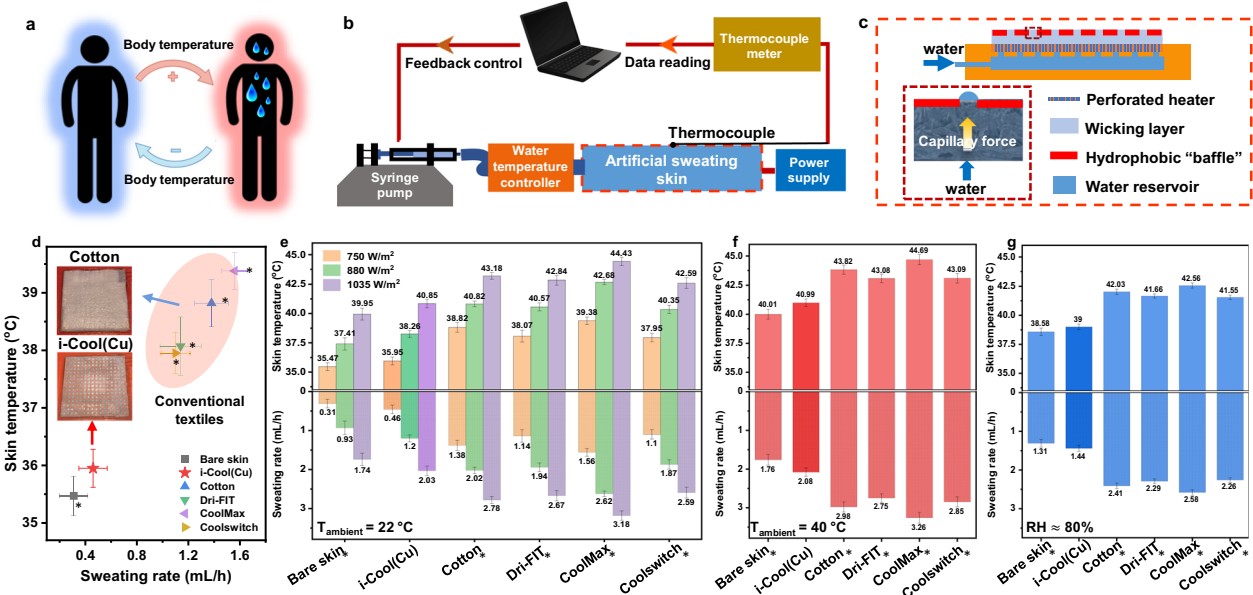

**Fig. 4 Artificial sweating skin platform with feedback control loop and measurements on it. a**, Schematic of human body temperature self-regulation mechanism. When body temperature increases, human body perspires to cool down its own temperature, which leads to reduction or suspension of perspiration in reverse. **b**, Schematic of the artificial sweating skin platform with feedback control loop simulating human body temperature self-regulation mechanism. **c**, Schematic of the detailed structure of the artificial sweating skin. The schematic in the red dash box shows the working mechanism of the modified Janus-type wicking layer which realizes uniform sweating mimicking human skin sweating scenario. **d**, Measurement results of skin temperature and sweating rate for bare skin, i-Cool (Cu) and commercial textiles (skin power density: 750 W/m², ambient temperature: 22 °C). Insets show the photographs of i-Cool (Cu) and cotton after one-hour stabilization during the tests. Asterisk, Statistical significance of skin temperature and sweating rate between the i-Cool (Cu) and other textiles, Welch's $t$-test $p < 0.001$. **e**, Measurement results of skin temperature and sweating rate for bare skin, i-Cool (Cu) and other conventional textiles under different skin power densities. Asterisk, Statistical significance of skin temperature and sweating rate between the i-Cool (Cu) and other textiles at 750 W/m², 880 W/m² and 1035 W/m², Welch's $t$-test $p < 0.001$. **f**, Measured skin temperature and sweating rate at high ambient temperature (40 °C). 750 W/m² power density was applied. Asterisk, Statistical significance of skin temperature and sweating rate between the i-Cool (Cu) and other textiles, Welch's $t$-test $p < 0.001$. **g**, Measured skin temperature and sweating rate in high relative humidity ambient (~80%). Asterisk, Statistical significance of skin temperature and sweating rate between the i-Cool (Cu) and other textiles, Welch's $t$-test $p < 0.001$. All the error bars represent standard deviation of measured data.

Note 6 and Supplementary Fig. 18 for more discussion for this measurement). All the measurements were performed from the same initial state. The skin temperature and sweating rate (i.e. water pumping rate) after stabilization were measured. Figure 4d shows the experimental results when skin power density was ~750 W/m² and ambient temperature was 22 °C. The cooling performance of i-Cool (Cu) is very similar to the bare skin, which is recognized as the most efficient cooling approach since sweat evaporation can directly take away heat from the skin. Compared to the conventional textiles, i-Cool (Cu) exhibited evidently lower skin temperature (~2.8 °C lower than cotton, ~2 °C temperature difference with Dri-FIT and Coolswitch, ~3.4 °C temperature difference with CoolMax). The sweating rate provided for the conventional textiles was over 2–3 times as much as i-Cool (Cu). It proves that conventional textiles cannot achieve better cooling effect even with much more available sweat. On the other hand, i-Cool (Cu) is able to unlock the cooling power of sweat more efficiently, which can deliver improved cooling effect with reduced sweating dehydration. As a result, conventional textiles would become highly wet after perspiration, whereas i-Cool (Cu) could retain a much drier state (insets of Fig. 4d), which is a comprehensive effect of evaporation ability and sweat evaporative cooling efficiency.

We tested the Cu heat conductive matrix and nylon 6 nanofibre film separately. The departure of the heat conduction component and water transport component makes both of them less efficient in evaporative cooling, as exhibited in Supplementary Fig. 19. These tests illustrate the key factor to achieve an

effective cooling effect is the integrated functional design of heat conduction and sweat transportation. Different cotton samples with various area mass density were also tested (See Supplementary Note 7 and Supplementary Fig. 20 for more details). In our experiments, the thinnest cotton sample (26.5 g/m²) that is too transparent to be practically used still exhibited around 1.5 °C higher skin temperature than the i-Cool (Cu) textile. These results further validate the superiority of the i-Cool structure that is an integrated one with both heat conduction and sweat transportation.

The artificial sweating tests under different skin power densities to simulate changed human body metabolic heat production were also conducted. As displayed in Fig. 4e, the enhanced cooling performance showing lower skin temperature and reduced sweating rate in comparison to conventional textiles was still true when different skin power densities were applied. It verifies the advantages of i-Cool in a wide range of heat production.

Besides, the evaluation of performance under diverse ambient environment conditions was performed, especially in high temperature circumstances and high relative humidity surroundings in which perspiration is more likely to happen. At the ambient temperature of 40 °C, the evaporative cooling performance of i-Cool (Cu) textile and the conventional textiles is shown in Fig. 4f. The cooling performance distinction between the i-Cool (Cu) and the conventional textiles was still very apparent. To take a step further, we decreased skin power density of the artificial sweating skin to make skin temperature lower

than ambient temperature to compare bare skin, i-Cool (Cu) and cotton, to see if the high thermal conductivity design in the i-Cool (Cu) will cause adverse effect for skin temperature. Consequently, skin temperature with the i-Cool (Cu) was almost the same as bare skin and showed better performance than cotton, as shown in Supplementary Fig. 21, indicating its evaporative cooling effect surpassed the opposing heat conduction from the ambient. In addition to high ambient temperature, we also investigated the performance of i-Cool (Cu) and other conventional textiles in a high relative humidity (RH) environment (Fig. 4g). As the relative humidity was raised, skin temperature with all the textile swatches rose correspondingly. Nevertheless, the skin temperature of the i-Cool (Cu) was still much lower than the conventional textiles.

Moreover, we performed measurements to see how the parameters in the functional structure design of i-Cool (Cu) influence its performance (See Supplementary Note 8 and Supplementary Fig. 22 for more details). The results provide additional guidelines for personal perspiration management textile design.

**i-Cool practical application demonstration**. To further study the cooling effect of the i-Cool textile on human body, we developed a thermal simulation considering the coupled heat transfer, moisture vapor and liquid water transfer processes based on the actual human body with complex structure and dynamic physiological responses (See Supplementary Note 9, Supplementary Dataset 1 and Supplementary Fig. 23 for more details)[49–51]. The simulation results show that the i-Cool textile with improved evaporation ability and sweat evaporative cooling efficiency can achieve temperature reduction in both the skin temperature and core temperature of the human body compared to that with conventional textiles (Supplementary Fig. 23), which further validates the potential of the i-Cool structure design in efficient evaporative cooling for the human body.

To bridge the gap between i-Cool (Cu) concept demonstration to practical use, we demonstrated the feasibility via fabricating the i-Cool textile based on commercial fabrics. Firstly, we verified the replacement of Cu matrix by polymer materials with heat conductive coatings. As shown in Supplementary Fig. 24, the i-Cool textiles using silver (Ag) coated polyester (PET) and nanoporous polyethylene (NanoPE) matrices exhibit almost the same performance as i-Cool (Cu) in the artificial sweating skin test (experimental parameters: same as Fig. 3d). Furthermore, we fabricated i-Cool textiles based on commercial knitted fabrics made of PET fibers. Here, we chose Dri-FIT and CoolMax which were already tested as control samples as the substrates. Figure 5a illustrates the fabrication process: holes were cut by laser cutting on the original fabric, after which it went through a facile electroless plating process. The Ag coating was deposited onto every fiber's surface of the fabric. Next, cellulose fibers were filled into the holes of the fabric, and prepared nylon 6 nanofibre film was transferred onto the fabric via press lamination to realize the i-Cool (Ag) textile which possessed the desired i-Cool structure. It is worthwhile to point out the fabrics we selected and the electroless plating method are not the only choices. Other textile material and other methods offering heat conductive coatings can be utilized. Alternatively, heat conductive fibers can be applied as well for the heat transport matrix. Figure 5b shows the photograph of the i-Cool (Ag) textile sample swatch (Dri-FIT as substrate). The photograph viewing from the i-Cool (Ag) bottom is exhibited in the inset of Fig. 5c, and the SEM images of the Ag coated PET fibers (Fig. 5c, Supplementary Fig. 25) show the Ag coating is conformal and uniform. The branched structure

formed in the electroless plating process can potentially enlarge evaporation area as well. The photograph and SEM images of i-Cool textile with CoolMax substrate are shown in Supplementary Figs. 26 and 27.

Successively, we performed the same steady-state evaporation test and artificial sweating skin test for the i-Cool (Ag) textile. In the steady-state evaporation test, the curves of i-Cool (Ag) plotted with curves of i-Cool (Cu), cotton and Dri-FIT (Fig. 5d, e) exhibited that i-Cool (Ag) exhibited very similar performance to the i-Cool (Cu) textile. Compared to the original Dri-FIT textile acting as the substrate, i-Cool (Ag) owns significantly improved evaporation performance and evaporative cooling efficiency, which is owing to the i-Cool functional structure. Also, in the artificial sweating skin test, i-Cool (Ag) and i-Cool (Cu) presented comparable cooling performance for personal perspiration management, which was significantly improved in contrast to cotton and Dri-FIT. This is also true for the i-Cool textile prepared with CoolMax substrate (Supplementary Fig. 28). With only sweat transportation channels, the modified Dri-FIT and CoolMax showed weaker cooling performance (Supplementary Fig. 28), which verifies the i-Cool structure combining heat conduction with water transportation provides superior strategy in personal perspiration management. These results demonstrate the feasibility of readily applying the i-Cool concept to practical usage.

In summary, we report a novel concept of i-Cool textile with unique functional structure design for personal perspiration management. The innovative employment of integrated water transport and heat conductive functional components together ensures not only its wicking ability, but also the fast evaporation rate, enhanced evaporative cooling effect and reduction of human body dehydration for human body via utilizing sweat in a highly efficient manner, which was demonstrated by the transient and steady-state evaporation test. An artificial sweating skin platform with feedback control loop simulating human body perspiration situation was realized, on which the i-Cool (Cu) textile shows comparable performance to the bare skin and apparent cooling effect with less provided sweat compared to the conventional textiles. Also, the structure advantage maintains under various conditions of exercise and ambient environment. Besides, the practical application feasibility of the i-Cool design principles was demonstrated, exhibiting decent performance. Therefore, we expect the i-Cool textile will open a new door and provide new insights for the textiles for personal perspiration management.

## Methods

**Textile preparation**. The Cu matrix used in the i-Cool (Cu) textile sample (main text) was prepared with Cu foil (~25 μm thickness, Pred Materials) laser cut via DPSS UV laser cutter. A pore array (2 mm diameter, 3 mm pitch) on the Cu foil was created to realize the Cu matrix. Nylon 6 nanofibre film was prepared by electrospinning. The nylon 6 solution system used in this work is 20 wt% nylon-6 (Sigma-Aldrich) in formic acid (Alfa Aesar). The polymer solution was loaded in a 5 mL syringe with a 22-gauge needle tip, which is connected to a voltage supply (ES30P-5W, Gamma High Voltage Research). The solution was pumped out of the needle tip using a syringe pump (Aladdin). The nanofibres were collected by a grounded copper foil (Pred Materials). The applied potential was 15 kV. The pumping rate was 0.1 mL/h. The distance between the needle tip and the collector is 20 cm. After collecting nylon 6 nanofibres of desired mass, the nylon 6 nanofibre film (~4.5 g/m², ~25 μm thickness) was transferred and laminated on the Cu matrix. A hydraulic press (MTI) was used to press nylon 6 nanofibres both into the holes and on the top of the Cu matrix. The fabricated i-Cool (Cu) was ~45 μm thick and 107.7 g/m². The varied parameters of the i-Cool (Cu) textile are shown in Supplementary Fig. 22. To fabricate the i-Cool (Ag) textile sample, same pore array as above was cut by laser cutter (Epilog Fusion M2 laser cutter) for the Dri-FIT or CoolMax textiles. Then, the fabric was cleaned and modified with polydopamine (PDA) coating for 2 h in an aqueous solution that consists of 2 g/L dopamine hydrochloride (Sigma Aldrich) and 10 mM Tris-buffer solution (pH 8.5, Teknova)[52]. For electroless plating of silver (Ag), the PDA-coated fabrics were then

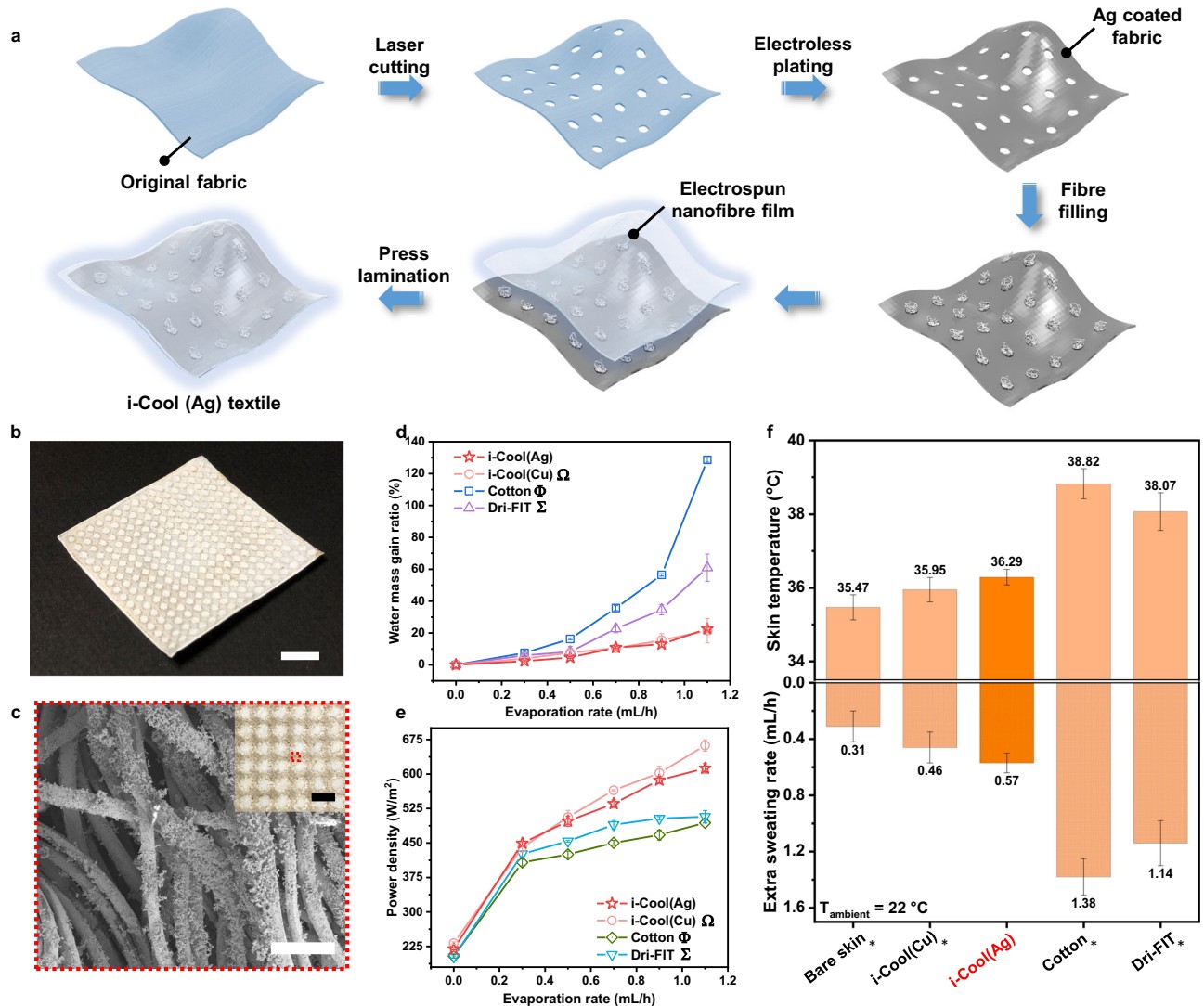

**Fig. 5 Practical application feasibility demonstration of the i-Cool functional structure via i-Cool (Ag) textile. a,** Illustration of the fabrication process of i-Cool (Ag) textile based on a commercially available fabric. **b,** Photograph of as-fabricated i-Cool (Ag) textile based on Dri-FIT as the substrate. Scale bar, 1 cm. **c,** SEM image showing the uniform and conformal Ag coating on the PET fibers of the fabric substrate. Scale bar, 50 μm. The inset shows the photograph of i-Cool (Ag) viewing from its bottom. Scale bar, 4 mm. **d,** Measured water mass gain ratio of i-Cool (Ag) and other textiles at different evaporation rate in the steady-state evaporation test. Omega symbol, Statistical significance between the i-Cool (Ag) and i-Cool (Cu), Welch's $t$-test $p <$ 0.1 at 0.3 mL/h and 0.5 mL/h, no statistical significance for others. Phi symbol, Statistical significance between the i-Cool (Ag) and cotton, Welch's test $p <$ 0.05 at 0.3 mL/h, $p <$ 0.001 for others. Sigma symbol, Statistical significance between the i-Cool (Ag) and Dri-FIT, Welch's test shows no statistical significance at 0.5 mL/h, $p <$ 0.05 at 0.7 mL/h, 1.1 mL/h, $p <$ 0.01 for others. **e,** Measured power density of i-Cool (Ag) and other textiles at different evaporation rate in the steady-state evaporation test. Omega symbol, Statistical significance between the i-Cool (Ag) and i-Cool (Cu), Welch's $t$-test $p <$ 0.01 at 0 mL/h, 1.1 mL/h, $p <$ 0.05 at 0.7 mL/h, no statistical significance for others. Phi symbol, Statistical significance between the i-Cool (Ag) and cotton, Welch's test $p <$ 0.001 at 0.7 mL/h and 0.9 mL/h, $p <$ 0.01 for others. Sigma symbol, Statistical significance between the i-Cool (Ag) and Dri-FIT, Welch's test $p <$ 0.05 at 0.5 mL/h, $p <$ 0.01 at 0.3 mL/h and 0.7 mL/h, $p <$ 0.001 for others. **f,** Measured skin temperature and sweating rate of the i-Cool (Ag) textile on the artificial sweating skin platform with feedback control loop. Asterisk, Statistical significance of skin temperature and sweating rate between the i-Cool (Ag) and other textiles, Welch's $t$-test $p <$ 0.001. All the error bars represent standard deviation of measured data.

dipped into a 25 g/L AgNO₃ solution (99.9%, Alfa Aesar) for 30 min to form the Ag seed layer. After rinsing with deionized (DI) water, the fabric was immersed into the plating bath solution containing 4.2 g L⁻¹ Ag(NH₃)₂⁺ (made by adding 28% NH₃·H₂O dropwise into 5 g L⁻¹ AgNO₃ until the solution became clear again) and 5 g L⁻¹ glucose (anhydrous, EMD Millipore Chemicals)[53] for 2 h. Next, the fabric was turned over and placed into a new plating bath for another 2 h. After drying, cellulose fibers were filled into the cut pores by extraction filtration of paper pulp. Then, nylon 6 nanofibre film (~2-2.5 g/m²) was added onto it by the same process described above. The as-prepared i-Cool (Ag) (based on Dri-FIT) is ~175 g/m². The one based on CoolMax is ~199 g/m². For i-Cool samples based on Ag-coated PET and NanoPE film matrix, the PET matrix (~50 μm thickness) and NanoPE matrix (~25 μm thickness) were prepared by laser cutting in the same way, and went through the same Ag coating process and nylon 6 nanofibre film

lamination. The cotton textile sample was from a common short-sleeve T-shirt (100% cotton, single jersey knit, 135 g/m², ~400 μm thickness, Dockers). The Dri-FIT textile sample was from a regular Dri-FIT T-shirt (100% PET, single jersey knit, 143 g/m², ~400 μm thickness, Nike). The CoolMax textile sample was from a T-shirt made of 100% CoolMax Extreme polyester fibers (100% PET, single jersey knit, 166 g/m², ~445 μm thickness, purchased from Galls.com). The Coolswitch textile sample was from a Coolswitch T-shirt (91%PET/9% Elastane, French terry knit, 140 g/m², ~350 μm thickness, Under Armour).

**Material characterization.** The optical microscope images were taken with an Olympus optical microscope. The SEM images were taken by a FEI XL30 Sirion SEM (5 kV) and a FEI Nova NanoSEM 450 (5 kV).

**Wicking rate measurement**. The wicking rate measurement method was based on AATCC 198 with modification. $5 \times 5$ cm textile samples were prepared ahead. 0.1 mL of distilled water was placed on the simulated skin platform by pipette. Then textile samples were covered on the water, and the time of water reaching the circle of 1.5 cm in radius on the top surface of textile was recorded. Wicking rate was calculated using wicking area divided by wicking time.

**One-way water transport characterization**. A $5 \times 5$ cm textile sample was fixed onto an acrylic frame that had a $4 \times 4$ cm square hole. Camera was placed right above the frame or underneath the frame to shoot videos. In total 20 µL of deionized water was added onto one side of textile sample and the water transport process was filmed. The testing time was calculated by an image processing software (SketchAndCalc Area Caculator). We calculated the $S_{inner}$, $S_{outer}$ and $\mu$ at the testing time of 15 s. $S_{inner}$ or $S_{outer}$ = water spreading area/testing time. $\mu$ is a one-way transport index, which is defined as $S_{outer}/S_{inner}$.

**Thermal resistance measurement**. The cut bar method adapted from ASTM 5470 was used to measure thermal resistance. In this setup, eight thermocouples are inserted into the center of two 1 inch × 1 inch copper reference bars to measure the temperature profiles along the top and bottom bar. A resistance heater generates a heat flux which flows through the top bar followed by the sample and then bottom bar after which the heat is dissipated into a large heat sink. The entire apparatus (top bar, sample, bottom bar) is wrapped in thermal insulation. A modest pressure of approximately 15 psi was applied at the top bar to reduce contact resistance, and no thermal grease was used due to the material porosity. The temperature profiles of the top and bottom copper bars are then used to determine both the heat flux and the temperature drop across the sample stack, which can derive the total thermal resistance ($R_{TOT}$). Plotting the $R_{TOT}$ versus the number of sample layers, the sample thermal resistance with contact thermal resistance between samples can be obtained from the slope of the line.

**Water vapor transmission property tests**. The upright cup testing procedure was based on ASTM E96 with modification. Medium bottles (100 mL; Fisher Scientific) were filled with 80 ml of distilled water, and sealed with the textile samples using open-top caps and silicone gaskets (Corning). The exposed area of the textile was 3 cm in diameter. The sealed bottles were placed into an environmental chamber in which the temperature was held at 35 °C and relative humidity was 30 ± 5%. The mass of the bottles and the samples was measured periodically. By dividing the reduced mass of the water by the exposed area of the bottle (3 cm in diameter), the water vapor transmission was calculated. The evaporative resistance measurement was based on ISO 11092/ASTM F1868 with modification. A heater was used to generate stable heat flux mimicking the skin. A metal foam soaked with water was placed on the heater. A waterproof but vapor permeable film was covered on the top of the metal foam to protect the textile sample from contact with water. The whole device was thermally guarded. For different textile samples, we adjusted the heat flux to maintain the same skin temperature (35 °C) for all measurements. The ambient temperature was controlled by the water recirculation system at 35 °C, and the relative humidity was within 24 ± 4%. The evaporative resistance was calculated by $R_{ef} = \frac{(P_s - P_a) \bullet A}{H} - R_{ebp}$, where $P_s$ is the water vapor pressure at the plate surface, which can be assumed as the saturation at the temperature of the surface, $P_a$ is the water vapor pressure in the air, $A$ is the area of the plate test section, $H$ is the power input, and $R_{ebp}$ is the value measured without any textile samples.

**Water vapor thermal measurement**. The artificial sweating skin platform was utilized in this measurement. A steady power density (580 W/m²) and water flow rate (0.25 mL/h) were adopted. An acrylic frame (thickness: 1.5 mm) with a crossing was laser cut and placed on the platform to support the textile samples avoiding the liquid water contact. Stable skin temperature was read. The ambient was 22 °C ± 0.2 °C, 40 ± 5% relative humidity.

**Transient droplet evaporation test**. The skin was simulated by a polyimide insulated flexible heater (McMaster-Carr, 25 cm²) which was connected to a power supply (Keithley 2400). A ribbon type hot junction thermocouple (~0.1 mm in diameter, K-type, Omega) was in contact with the top surface of the simulated skin to measure the skin temperature. The heater was set on a 10 cm-thick foam for heat insulation. During the tests, water (37 °C) was added onto the simulated skin and textile samples were covered on the simulated skin immediately. The skin temperatures with wet textile samples during water evaporation were measured with an assorted combination of initial water amount and generated area power density of simulated skin. The average evaporation rate was calculated by dividing the initial water amount by evaporation time. The end point of the evaporation was defined as the inflection point between the relatively stable range and the rapid increase stage of temperature. The average skin temperature referred to the average temperature reading spanned the evaporation stage in which skin temperature was relatively stable. The mass of wet textile samples was measured by a digital balance (U. S. Solid, 0.001g accuracy) to track the water mass loss during the evaporation.

The tests were all performed in an environment of 22 ± 0.2 °C, 40 ± 5% relative humidity.

**Steady-state evaporation test**. The skin was simulated by a polyimide insulated flexible heater (McMaster-Carr, 25 cm²) which was connected to a power supply (Keithley 2400). It was covered by a 1.5 mm-thick acrylic board with grooves made by laser cutting (Epilog Fusion M2 laser cutter) on its top surface. A ribbon type hot junction thermocouple (~0.1 mm in diameter, K-type, Omega) was sealed in a groove by PDMS to measure the skin temperature. A needle connected to a tube and a syringe pump (Harvard, PHD 2000) was also sealed in one groove of the acrylic board, but with head exposed for water outage. The heater was set on a 10 cm-thick foam for heat insulation. During the tests, water in the tube was heated by a proportional–integral–derivative (PID) temperature controller (Omega Engineering) at 37 °C before flowing onto the artificial skin. Textile samples were placed on the artificial skin surface. The applied power density was adjusted to let measured skin temperature fluctuate around 35 °C. After stabilization for a period of time, the mass of wet textile samples was measured by a digital balance (U. S. Solid, 0.001g accuracy), and power density was recorded. The tests were all performed in an environment of 19.5 ± 0.3 °C, 35 ± 5% relative humidity.

**Fabrication of Janus-type wicking layer with limited water outlets**. A filter paper (Qualitative, Whatman) was used as the wicking layer. An acrylic board was laser cut into a mask with Epilog Fusion M2 Laser and placed on the top of the filter paper. Polydimethylsiloxane (PDMS) base and curing agent (Sylgard 184, Dow Corning) with mass ratio of 10:1 were dispersed into hexane (Fisher Scientific) with volume ratio 1:10. The PDMS solution was sprayed onto the masked filter paper that was on a heating plate, which helped with faster volatilization of hexane. After drying and curing, the PDMS formed hydrophobic coating layer only on the uncovered place of the top surface of the filter paper, which could absorb and transport water from the bottom surface but provide limited water outlets on the top surface.

**Artificial sweating skin test with feedback control loop**. The water reservoir (5 cm × 5 cm × 2.5 mm) with water inlet (whole part size: 8 cm × 8 cm × 3.5 mm) was made by 3D printing (FlashForge Creator Pro). A cover with a 9 × 9 hole (diameter: 3 mm) array (hole array area: 5 cm × 5 cm, whole part size: 8 cm × 8 cm × 1.5 mm) was also 3D printed and bound with the water reservoir part. The water reservoir was connected to a syringe pump (Harvard, PHD 2000). The pumped water was heated at 37 °C by a heater (Omega Engineering) and a proportional–integral–derivative (PID) temperature controller (Omega Engineering). A polyimide insulated flexible heater (McMaster-Carr, 25 cm²) with laser cut water outlets was adhered to the holey cover. The heater was connected to a power supply (Keithley 2400). Then, the fabricated Janus-type wicking layer with limited water outlets was attached to the heater layer to serve as the skin surface. A ribbon type hot junction thermocouple (~0.1 mm in diameter, K-type, Omega) connected to a thermocouple meter (Omega Engineering) was in contact with the top surface of the Janus-type wicking layer to measure the skin temperature. The thermocouple meter, syringe pump and power supply were all controlled by a LabView program, which can alter the pumping rate (extra sweating rate) according to the thermometer reading (skin temperature) in real time. Before the test, the artificial sweating skin platform was filled with water in advance. The perspiration threshold skin temperature was set to be 34.5 °C, over which the sweating rate was linearly dependent on skin temperature[47,48]. The relationship between pumping rate and skin temperature was set as pumping rate (mL/h) = 0.32*skin temperature (°C) −11.04, which was decided according to previous research and reasonable human body perspiration rate range. The whole set-up was in a space without forced convection. No chamber with cover for the set-up was used to avoid water vapor accumulation except the high-humidity test. In the high-humidity test, a humidifier was placed next to the testing platform and they are enclosed together to change the humidity. The initial air temperature in the chamber was 22 °C but about 1–2 °C reading variation of the ambient temperature thermometer was observed, perhaps due to the water vapor condensation, but no obvious influence on the skin temperature was observed. In other cases, if no ambient temperature and relative humidity are specified, the ambient temperature was 22 ± 0.2 °C and ambient relative humidity was 40 ± 5%.

## Data availability
The data that support the findings of this study are available from the corresponding author upon reasonable request.

## Code availability
The code for thermal simulation of actual human body is available from the corresponding author upon reasonable request.

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

## Acknowledgements

We acknowledge the great help of P. Zhu, C. Lau, G. Gerboni, Z. Yu, and Y. Zheng. Part of this work was performed at the Stanford Nano Shared Facilities and the Stanford Nanofabrication Facility. J.S., C.D., and R.P. acknowledge the support of the Laboratory Directed Research and Development Program (LDRD) at Lawrence Berkeley National Laboratory under contract # DE-AC02-05CH11231.

## Author contributions

Y.C. and Y.P. conceived the idea. Y.P. designed and conducted the experiments. Y.P., W.L., and B.L. performed the feedback control loop construction and programming. W.L. and W.J. conducted the simulation. B.L. drew the schematics. J.T. and G.Z. helped with sample preparation. J.S and J.Z. performed the thermal resistance measurement. G.W. helped with statistical analysis. Y.Z. and C.Z. helped with laser cutting process. W.H. and T.W. provided helpful discussion. Y.C. and R.P., C.D., S.F., K.G. supervised the project. All the authors provided helpful discussion on this project and contributed to manuscript writing.

## Competing interests

The authors declare no competing interests.
