## [Peer Review File · Nature Communications]

Reviewers' Comments:

Reviewer #1:

Remarks to the Author:

Dear Editor,

Sorry for late submission of the review as I were travelling on the airplane yesterday and I had to go through all the COVID-19 tests and waiting for results for many hours, which was very exhausting.

After reviewing the revised manuscript, reply to reviewers' comments and supporting materials, I am glad to see the manuscript has been improved substantially. The authors have made detailed and comprehensive responses to Reviewers' comments and incorporated the comments into their revised manuscript and the Extended Data document. I am happy to recommend the manuscript to be accepted for publication in Nature Communication.

Response to Referees' comments (NCOMMS-21-18949-A):

Referee #1 (Remarks to the Author):

Dear Editor,

Sorry for late submission of the review as I were travelling on the airplane yesterday and I had to go through all the COVID-19 tests and waiting for results for many hours, which was very exhausting.

After reviewing the revised manuscript, reply to reviewers' comments and supporting materials, I am glad to see the manuscript has been improved substantially. The authors have made detailed and comprehensive responses to Reviewers' comments and incorporated the comments into their revised manuscript and the Extended Data document. I am happy to recommend the manuscript to be accepted for publication in Nature Communication.

Response:

We greatly appreciate the reviewer's comments on our manuscript and recommendation for publication in Nature Communication. We are also very grateful to reviewers for investing valuable time in reviewing our manuscript and giving us constructive suggestions!